# OFFLINE-ONLINE REINFORCEMENT LEARNING: EXTENDING BATCH AND ONLINE RL

## ABSTRACT

Batch RL has seen a surge in popularity and is applicable in many practical scenarios where past data is available. Unfortunately, the performance of batch RL agents is limited in both theory and practice without strong assumptions on the data-collection process e.g. sufficient coverage or a good policy. To enable better performance, we investigate the offline-online setting: the agent has access to a batch of data to train on but is also allowed to learn during the evaluation phase in an online manner. This is an extension to batch RL, allowing the agent to adapt to new situations without having to precommit to a policy. In our experiments, we find that standard RL agents trained in an offline-online manner can outperform agents trained only offline or online, sometimes by a large margin, highlighting the potential of this new setting.

## 1 INTRODUCTION

The batch RL setting has attracted much attention in recent times Fujimoto et al. (2019b); Wu et al. (2021); Zhang et al. (2021; 2020). In this problem, the agent has to learn a good policy from a fixed batch of data, without acting in the environment (Ernst et al., 2005), making it a good fit for many practical applications where only past logged data is available.

This problem setting has proven to be difficult. Empirically, in the case where a behaviour policy has collected the data, batch RL algorithms are only able to provide a policy matching or slightly exceeding the original behaviour policy's performance (Fujimoto et al., 2019a; Kumar et al., 2019). From the theoretical side, lower bounds exist showing that without strong assumptions on the data distribution and features, it may take exponentially many samples to learn a good policy (Zanette, 2020; Chen & Jiang, 2019). Otherwise it seems an algorithm can only produce the best policy which is well-covered by state-actions contained within the batch (Liu et al., 2020).

These findings contrast results in the online setting, where there exist many effective algorithms both in practice (Haarnoja et al., 2018; Mnih et al., 2015) and in theory (Jin et al., 2020a). As such, it is natural to consider a variant of the batch setting, closer to online learning. In the standard batch RL problem, the learner observes a pre-collected dataset and outputs a policy, which is evaluated by running it on the environment. The main difficulty is the lack of feedback from the environment. As such, the learner is unable to assess the quality of a policy it proposes.

We may instead consider a setting where the agent is allowed to do further learning during the evaluation phase, which we call the *offline-online RL* setting. While a natural extension to batch RL, this setting remains largely unexplored. In the offline-online setting, the goal is to learn an effective policy as quickly as possible. While this may involve starting with a good policy learned offline as in batch RL, the agent now has the chance to correct its policy by observing new transitions. This can help avoid a major roadblock to offline RL—extrapolation error (Levine et al., 2020; Fujimoto et al., 2019b). This issue arises due to the agent encountering states not found in the dataset when it is evaluated. Oftentimes, the agent may choose bad actions in these new states, without a chance to learn from its mistakes. This implicitly limits the agent's performance based on which states are present in the dataset. In many applications, this problem is unavoidable since we cannot expect to collect a batch of data covering all possible situations. In the offline-online setting, the agent sidesteps this issue as it can improve its policy at new states.

Compared to online RL, the offline-online agent now has extra information in the form of this dataset. As such, the agent can be expected to perform better since we can jumpstart the policy to higher-level of performance through batch RL. This may be especially desirable in practical applications where one would require the initial deployed policy to perform well. In these settings, starting tabula rasa, with online RL may be an unacceptable solution. From a theoretical standpoint, it has been shown that online learning performance can be improved (Xie et al., 2021) when the agent has access to a good reference policy, a similar setting.

In this paper, we study the difference between the online, batch and offline-online RL settings from an empirical perspective. We look at the simplest instance of offline-online RL where we transfer a policy (value function) and a learned representation from the offline phase to the online phase. While we can expect algorithms to perform better in the offline-online setting compared to batch RL as we allow additional learning, as far as we know, no work has delved deeper into the differences. In offline-online RL, the goal is to use the batch of data to learn faster online, similar to the goal of representation learning: learn features that facilitate effective linear updates. However, the representation learned offline will also be limited, and so should continue to be adapted online too. We show that an approach, called Two-Timescale Networks (TTNs) (Chung et al., 2018), that separate representation learning and value learning, are well-suited to this offline-online setting. To the best of our knowledge, there are no existing approaches that exploit this offline-online setting to learn a representation and policy offline, for further updating online. The batch RL setting focuses only on learning a policy that is fixed and deployed online. Existing works often focuses on unsupervised RL (Schwarzer et al., 2021; Ghosh & Bellemare, 2020), where a representation is learned without explicit rewards but the agent can control the data gathering policy.

As a part of this work, we show that there is (a) a significant benefit in comparison with offline methods to allow the policy to continue to adapt online and (b) a significant benefit to use the offline batch in comparison to purely online methods. The offline setting is often characterized by the need for safety, and concerns with allowing the policy to adjust. This work, however, highlights the large opportunity cost with doing so, and suggests that a more suitable path is to consider even small adaptation, such as has been considered for high-confidence policy improvement. On the flip side, it also highlights that it is restrictive to only learn online from scratch; if data is available (as it often is), then this can be leveraged to significantly speed learning both the representation and policy.

Further experiments in batch RL measuring the performance to the size of the dataset, the data-collection policy and the amount of training done. Here, we find that larger datasets and a good policy can be critical to good results. There are no clear trends for the impact of the amount of training. In each environment, different amounts of training are best. Taken together, these experiments highlight the difficulty of batch RL, where many different factors have to be correct to obtain reasonable results.

## 1.1 RELATED WORK

There has been quite a bit of work considering transferring learning from one setting to another; here, that transfer occurs from learning in an offline data to then learn further online. However, most transfer settings involve transfering only the policy or transferring between related environments, where the agent can control data collection in both environments. There is however some work that has considered transferring objects other than a policy between the offline and online (evaluation) phases. One work has considered transferring options, where options (Sutton et al., 1999) are learned in the offline phase and then leveraged to speed up learning during the online phase (Ajay et al., 2021). Other directions of research are of a similar flavor, such as unsupervised representation learning, in which the agent seeks to learn a useful transformation of the observations for control without access to rewards (Schwarzer et al., 2021).

From batch RL, other examples include learning world models (Yu et al., 2020) for planning or learning uncertainty estimates (Osband et al., 2018; Jaques et al., 2019) to induce either optimism Lai & Robbins (1985) or pessimism (Jin et al., 2020b; Buckman et al., 2020)in the online phase, either of which may be useful in certain situations. Different problems can also be viewed as part of the offline-online setting such as inverse RL (Ng et al., 2000), where we seek to learn the appropriate reward function on a batch of data to optimize during the online phase. Offline-online RL is also related to safe RL (Thomas et al., 2015; Kakade & Langford, 2002) where the agent uses a fixed-policy for some time to collect enough data to make a policy improvement step with high-confidence.

From the perspective of offline-online RL, we can imagine other extensions of algorithms designed for online RL, such as using an offline dataset to jumpstart the meta-learning process of update rules (Javed & White, 2019) or hyperparameters (Xu et al., 2018). Through these connections, we see that offline-online RL is a natural setting to study, suggesting new algorithms and relevant in practical situations.

## 2 BACKGROUND

We model the environment as an MDP $\mathcal{M} = (\mathcal{S}, \mathcal{A}, \mathcal{R}, \mathcal{P}, \gamma)$ where $\mathcal{S}$ are the states, $\mathcal{A}$ are the actions, $\mathcal{R} : \mathcal{S} \times \mathcal{A} \to \mathbb{R}$ is the reward function, $\mathcal{P} : \mathcal{S} \times \mathcal{A} \to [0, 1]$ denotes the transition probabilities, and $\gamma \in [0, 1]$ is the discount factor.

*Online RL* (Sutton & Barto, 2018; Lai & Robbins, 1985). In this setting, the agent directly interacts with the environment. At each timestep, the agent chooses an action $a_t$ and the environment returns the next state $s_{t+1}$ and reward $r_{t+1}$. The agent can adapt to new data and change its policy after every single transition. For an episodic task, the goal is to accumulate the highest amount of return in each episode and eventually obtain the optimal policy.

*Batch RL* (Ernst et al., 2005; Fujimoto et al., 2019b; Levine et al., 2020). The agent receives a dataset $\mathcal{D} = \{(s_i, a_i, r_i, s_i')\}_{i=1}^{N}$, consisting of $N$ transitions from an environment $\mathcal{M}$. These transitions can be collected arbitrarily although they are often obtained by running a policy and recording the transitions observed by the agent. The goal of the agent is to use this dataset to learn a policy that performs well in the underlying environment. A key aspect is that once the algorithm has output a policy, it can no longer change it.

## 3 OFFLINE-ONLINE RL

In this section, we outline the proposed offline-online setting. There are two phases: The offline phase and the evaluation (online) phase. In standard batch RL, given a dataset $\mathcal{D}$, the agent would learn a policy $\pi$ in the offline phase. Then, during evaluation, this policy is fixed and deployed in the environment to assess its performance. Likewise, in the basic offline-online setting, a policy $\pi$ is learned in the offline phase. The difference is that in the evaluation phase, the policy is allowed to adapt online as it interacts with the environment. For a generic RL algorithm that has parameters $\theta$ and associated policy $\pi_\theta$ and an episodic task, we have the following pseudocode in Algorithm 1.

---

**Algorithm 1** Offline-Online RL

---

1: **procedure** OFFLINE-ONLINE RL($\theta, \mathcal{D}$)
2:     Given dataset $\mathcal{D}$ of transitions $(s, a, r, s')$
3:     Initialize agent parameters $\theta$
4:     **for** number of updates **do**
5:         Update $\theta$ using data from $\mathcal{D}$
6:     **end for**
7:     **for** $N$ number of online episodes **do**                      ▷ Evaluate the agent
8:         **while** episode not done **do**
9:             $a \leftarrow$ action chosen by $\pi_\theta$ given $s$
10:             $r, s' \leftarrow$ Environment(s, a)                      ▷ Get reward and next state
11:             $D \leftarrow (s, a, r, s')$                      ▷ Add sample to the batch data
12:             Update $\theta$ using $(s, a, r, s')$ (and possibly past samples)  ▷ Allow additional learning
13:             If episode ends, record the total return.
14:         **end while**
15:     **end for**
16: **end procedure**

---

Compared to online RL, the agent now has extra information in the form of this dataset. Thus, it could be expected that the agent perform better in this setting, with the amount of improvement depending on properties of the offline dataset given to the agent.

In this paper, we consider some simple approaches to offline-online RL where a learned value function and a representation are directly transferred from the offline phase and the online phase. As we will see, despite the simplicity, there are already major benefits to the approach.

To give a concrete example of how to set up an algorithm in this setting, we present the pseudocode for an offline-online variant of Two-Timescale Networks (TTNs), in Algorithm 2.

The TTN algorithm splits representation learning and value learning. For the representation, we use a neural network and treat the final hidden layer as a learned representation. This neural network is trained by using SGD on a surrogate loss, which would be minimizing the squared TD-error in this case (with a stop gradient on the target value). For value learning, we treat the representation as fixed and use fitted Q-iteration (FQI) (Ernst et al., 2005) tailored for linear function approximation, which solves for the parameters using the replay buffer. This may be an advantage since the updates can be larger and takes into account all available data at once, unlike SGD approaches, which must make small incremental updates.

---

**Algorithm 2** Offline-Online TTN

1: **procedure** TRAIN( $w, \theta, \hat{w}, \pi$ )
2:     Initialize $\theta, \hat{w}$ with Xavier initialization, $w$ to 0, $\tau$ to number of updates, and starting state $s$ according to the environment.
3:     Initialize data set $D$ from a batch of data generated by an arbitrary policy.
4:     **while** $|\theta_{t+1} - \theta_t| \geq \epsilon$ or $t \leq \tau$ **do**         ▷ **Offline-step**
5:         $\theta, \hat{w} \leftarrow$ Do 10000 gradient descent on feature learning using sample $(s, a, r, s')$ from mini-batch data $d \in D$
6:         $w \leftarrow$ Do 1 FQI update on value learning using sample $(s, a, r, s')$ from batch data $D$
7:         $t \leftarrow t + 1$
8:     **end while**
9:     **for** $N$ number of episodes **do**
10:         $a \leftarrow$ action chosen by $\pi$ given $s$
11:         $r, s' \leftarrow$ Environment(s, a)         ▷ Get reward and next state
12:         $D \leftarrow (s, a, r, s')$         ▷ Add sample to the batch data
13:         $\theta, \hat{w} \leftarrow$ Do SGD on feature learning using sample $(s, a, r, s')$ from mini-batch data $d \in D$
14:         **if** number of steps == 1000 **then**
15:             $w \leftarrow$ Do FQI update on value learning using sample $(s, a, r, s')$ from batch data $D$
16:         **end if**
17:     **end for**
18: **end procedure**

---

The difference with (original) online TTN is that, here, the agent first gets to learn on the given batch of data. This is done by doing some number of SGD updates from the dataset to learn the features, followed by multiple iterations of Fitted Q-iteration to learn the value estimates. In this way, offline-online TTN has access to both a better representation and a value function (policy) when starting the online phase. Compared to offline TTN, offline-online TTN is allowed additional updates after the offline phase. In doing so, the agent can adapt to any new data it observes in the online phase and can continue to improve the policy and representation. The complete pseudocode for offline and online TTN and DQN can be found in appendix A.2.2.

## 4 EXPERIMENTS

In the following sections, we investigate various aspects of the offline-online setting and compare to the more traditional batch and online RL settings. As a first experiment, we focus on the batch RL setting and look at offline-online DQN (Mnih et al., 2015), an agent which gets to train first on a batch of data and then perform online updates. We compare its performance to a representative batch RL algorithm, discrete BCQ (Fujimoto et al., 2019a). The batch of data is collected from a policy which performs well on the task and consists of 10 thousand transitions. Offline-online Two-Timescale Networks (TTN) is included as an agent that separates the representation learning and value learning processes (Chung et al., 2018) as an alternative approach.

For all experiments, we use 30 runs of each algorithm in the Mountain Car, Acrobot, Cartpole (Brockman et al., 2016), and Catcher (Tasfi, 2016) environments. To tune the learning rate, $\alpha$, and the regularization coefficients, $\lambda$ for TTN, we run the algorithms using grid search with the sweeps $\alpha \in \{10^{-1.0}, 10^{-2.0}, 10^{-2.5}, 10^{-3.0}, 10^{-3.5}, 10^{-3.75}, 10^{-4.0}\}$ and $\lambda \in \{0, 0.0003, 0.001, 0.002, 0.05, 0.01, 1, 2, 3, 5, 8\}$. To find a suitable number of updates for the offline phase, the offline-TTN algorithm was assessed for $\{70, 100, 150, 200\}$ number of updates. Afterwards, the number with the best performance is selected for the offline-online setting as well.

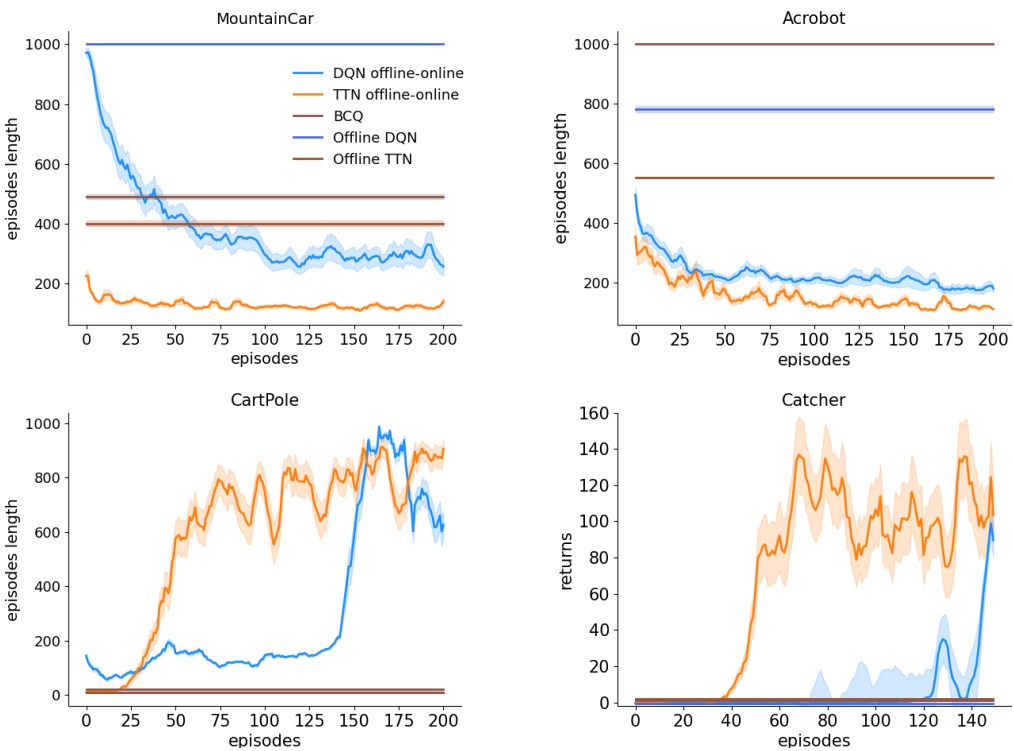

Figure 1: We compare the offline-online setting to batch RL. We see that while training only offline does not perform well in any environment alone, allowing learning afterwards can significantly improve the performance of the policy.

In Fig. 1 we see that the offline-online algorithms outperforms the batch RL agents, discrete BCQ (Fujimoto et al., 2019a) and offline DQN/TTN, by a significant margin. In Mountain Car and Acrobot, lower episode lengths are better while in Cartpole and Catcher, longer episode lengths are better. The learning curves are smoothed by averaging over a window of five episodes. Other experimental details can be found in the appendix.

In CartPole and Catcher, we see that the batch RL agents are unable to learn a policy that achieves non-neglible rewards. In these cases, we see that adding online training steps enables the agent to achieve a decent level of performance after some time. This highlights the importance of letting the agent learn during the evaluation phase, after it is trained on the offline batch. Looking at other settings, even though offline TTN can achieve a good performance in mountain car, additional training in the online phase still improves performance. In all cases, the performance of the offline-online agent is never worse than only offline training.

Next, we turn to the online RL setting and compare offline-online DQN and TTN with their purely online versions. As an additional comparison, we consider using the offline batch of data to initialize the replay buffer of the algorithms. We can view this is as a simple attempt to augment the online algorithms with an available batch of data and expect this to improve the performance of the online agent.

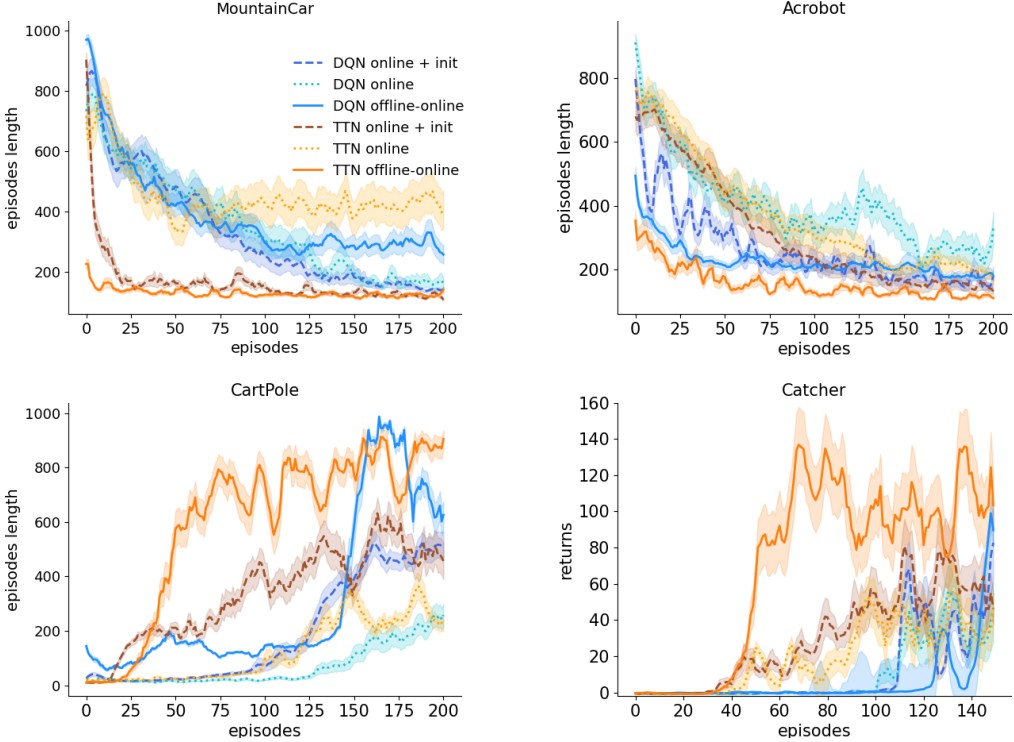

Figure 2: We compare the offline-online setting to purely online learning. We plot learning curves for standard online DQN/TTN, online DQN/TTN initialized with the offline dataset and offline-online DQN/TTN. We find that initializing with the offline dataset is better than the standard procedure but additionally training on it is still best.

Fig. 2 shows the learning curves for this setting. We see the offline-online agents improve on the purely online agent in every setting. Training offline before the online phase gives a jump-start in terms of the performance at the beginning and, in some cases, this advantage is retained throughout the entirety of training. The buffer initialization strategy also gives better performance across the board compared to the standard online agent and, in Mountain car and Acrobot, leads to a final performance similar to that of the offline-online agents' even if the performance is inferior in the early stages.

These two experiments confirms that the offline-online setting can produce better performing agents than either setting alone. Since this setting can naturally occur in many practical situations where some data is available beforehand, offline-online RL deserves further study. Specifically, compared to batch RL, there may be no need to restrict the agent to precommit to a policy after training on the dataset. We see that allowing adaptation to incoming data during deployment may yield significant benefits.

## 4.1 LEARNING A REPRESENTATION SEPARATELY

We investigate separating representation learning as in Two-Timescale Networks (TTN) or whether end-to-end learning is a better choice as is traditionally done.

The TTN algorithm splits representation learning and value learning. For the representation, we use a neural network and treat the final hidden layer as a learned representation. This neural network is trained by using SGD on a surrogate loss, which would be minimizing the squared TD-error in this case (with a stop gradient on the target value). For value learning, we treat the representation as fixed and use fitted Q-iteration (FQI) (Ernst et al., 2005), which solves for the parameters using the replay buffer as a batch of data. Since FQI is fairly computationally expensive, we only do this every 1000 steps. See the appendix for more details.

Revisiting figures 1 and 2, we can focus on comparing the performance of TTN and DQN. We find that TTN achieves better performance than DQN in the majority of cases. Notably, TTN seems to produce better solutions in the batch RL setting such as MountainCar, where offline DQN is not able to complete the task (it hits the timeout limit) but offline TTN is able to do so. In the online setting, from Fig. 2, again we have that TTN seems to have better performance overall, with a larger difference near the beginning of the online phase. As a tradeoff, since TTN uses FQI (Ernst et al., 2005) to solve for the next set of linear weights to approximate the value function in addition to SGD steps to learn the representation, the computational cost of TTN is higher in practice than than of DQN although the asymptotic cost is of the same order. These results suggest that end-to-end training may not be necessary to achieve good performance on these tasks and, in fact, separating the representation can be superior.

## 4.2 CHOICES OF OFFLINE DATASET

In this section, we perform certain ablation studies on the offline dataset. We can expect the properties of the dataset to influence the performance of the agent. For the offline-online setting, we test the impact of the size of the offline batch of data with DQN and TTN. The results are shown in Fig. 3.

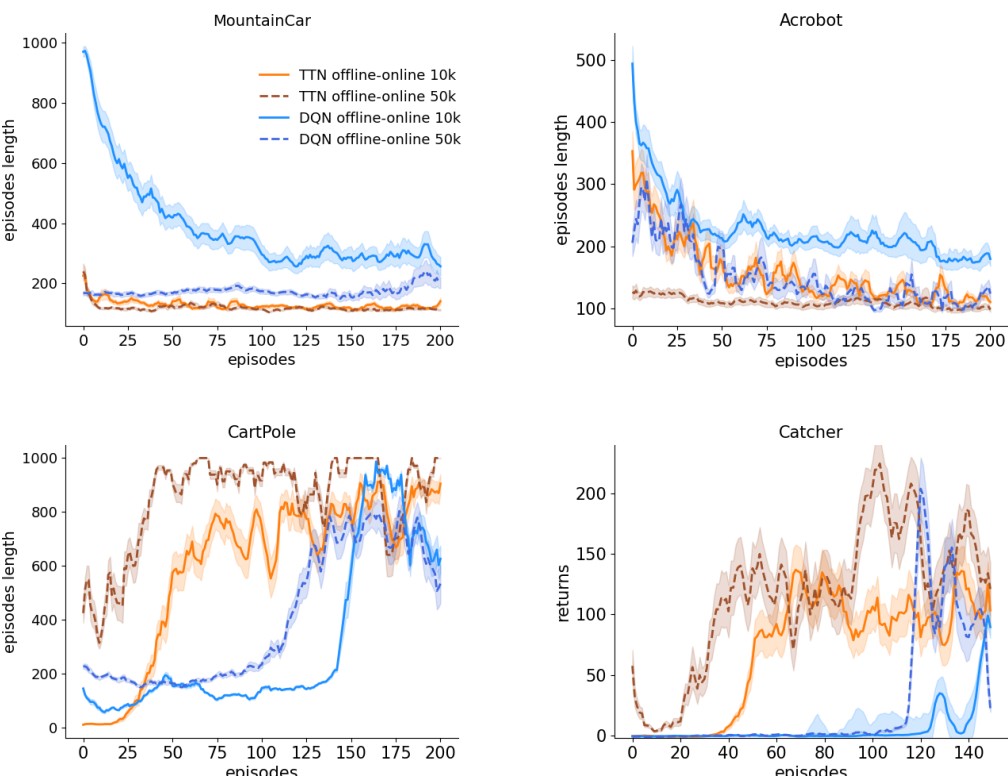

Figure 3: We present learning curves for two dataset sizes in the offline-online setting, 10 thousand and 50 thousand transitions for DQN and TTN. The larger dataset provides a noticeable boost.

From these plots, we see that the increasing the amount of data improves performance as expected. In Mountain Car, the difference is more remarkable as the larger dataset enables DQN to start the online phase with a good policy. This may be due to better coverage of the state space which makes offline training more successful.

We repeat the experiments comparing the offline-online setting to batch RL (Fig. 1) with the larger dataset of 50 thousand transitions. The results are qualitatively similar with the larger dataset although the performance of all the algorithms is improved, particularly the offline algorithms.

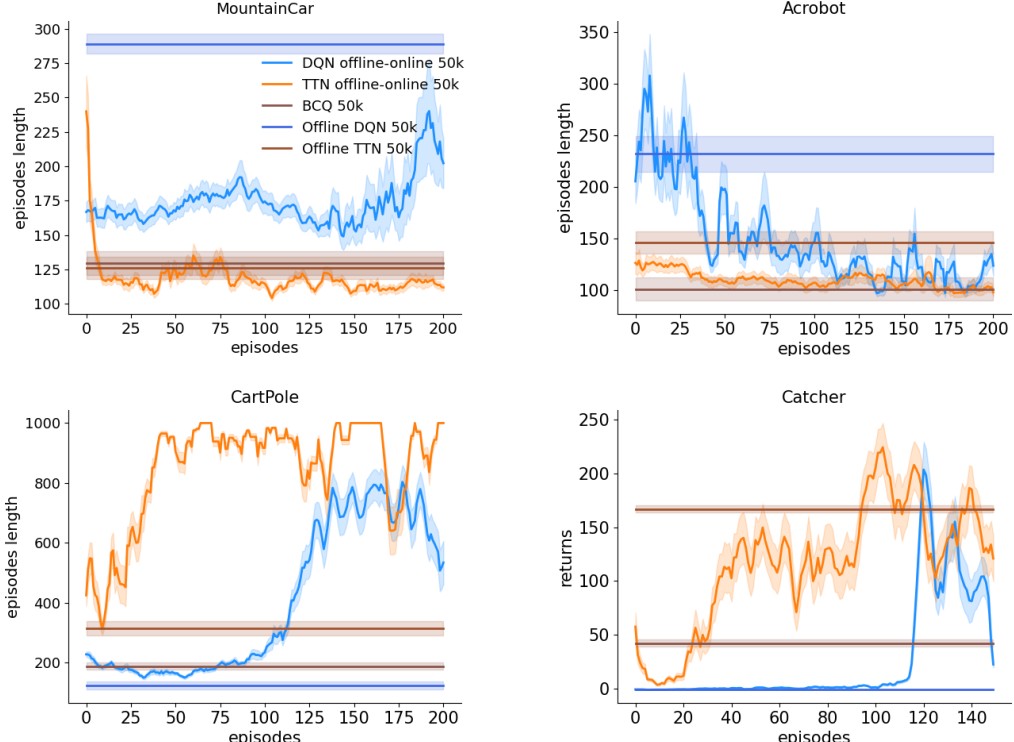

Figure 4: We compare the offline-online setting to batch RL when a dataset of 50 thousand transitions is used. We observe qualitatively similar behaviour as with 10 thousand transitions.

Furthermore, we provide more experiments for the batch RL setting. First, we test two sizes of datasets and two different data-collection strategies: using a near-optimal policy or a random policy with dataset sizes of 10 thousand or 50 thousand transitions. These are summarized in table 1.

We see that both the quality of the policy and the size of the dataset are important factors to the final performance of the agent. The data-collection policy impacts the results greatly. Often, the random policy does not seem to provide a dataset sufficient to achieve nontrivial performance (e.g. Mountain Car or Acrobot). Looking at TTN on Catcher or DQN on CartPole, we see that the size of the dataset can also be the difference between trivial and decent results. In general, we see that TTN performs better in the batch RL setting with performance always equal or better than DQN's. We hypothesize this may be due to fitted Q-iteration operating on the entire dataset at once for each update instead of stochastic updates as in DQN.

Finally, we investigate offline TTN further and look at how the amount of offline training impacts the performance. Since this setting resembles supervised learning, we may expect a similar trend where performance increases at first and then decreases due to "overfitting".

The results in Table 2 suggest that the previous hypothesis is too simplistic since we do not show any clear trends across environments. In Catcher, performance tends to increase with additional training but, in CartPole, it is the opposite. In Acrobot, the best performance is achieved by either a low number or a large number iterations, with the intermediate numbers producing worse results. The ideal amount of training seems to be highly specific to the environment and it may be difficult to tune for batch RL.

## 5  DISCUSSION

While batch RL is an appealing setting due to its relevance in many applications, we argue that allowing learning during the evaluation phase—the offline-online setting—may be a more natural

| Mountain Car | | DQN | | TTN | |
|---|---|---|---|---|---|
| | Size/Policy | Good | Random | Good | Random |
| | 10k | 1000 (0) | 1000 (0) | 401 (8.6) | 1000 (0) |
| | 50k | 289 (7.1) | 1000 (0) | 126 (8.1) | 1000 (0) |
| CartPole | | DQN | | TTN | |
| | Size/Policy | Good | Random | Good | Random |
| | 10k | 9.4 (1.5) | 9.9 (1.7) | 10.2 (0.9) | 9.5 (0.8) |
| | 50k | 121.6 (13.2) | 10 (1.0) | 313.5 (24.8) | 106.3 (7.1) |
| Acrobot | | DQN | | TTN | |
| | Size/Policy | Good | Random | Good | Random |
| | 10k | 782 (10.8) | 1000 (0) | 553 (2.0) | 999.5 (2.1) |
| | 50k | 232 (17.4) | 905 (7.8) | 146 (11.1) | 999.3 (2.1) |
| Catcher | | DQN | | TTN | |
| | Size/Policy | Good | Random | Good | Random |
| | 10k | -0.78 (0.021) | -0.73 (0.031) | 1.89 (0.19) | 0.87 (0.024) |
| | 50k | -0.73 (0.032) | -0.73 (0.029) | 167 (3.2) | 16.6 (1.9) |

Table 1: This table contains the results for training on an offline batch of data examining two factors: the size of the dataset (10 thousand or 50 thousand) and the quality of the policy (a good policy or a random policy). The experiments are done for both DQN and TTN. The number in the table is the average of 30 runs with one standard error indicated in parentheses. Entries highlighted in green show settings where nontrivial performance was achieved.

| Iterations | MountainCar | CartPole | Acrobot | Catcher |
|---|---|---|---|---|
| 70 | 201 (7.4) | **313.5 (24.8)** | 249 (30.7) | 23.9 (1.3) |
| 100 | **126 (8.1)** | 236.2 (16.7) | **146 (11.1)** | 25.6 (1.2) |
| 150 | 642 (12.0) | 135.8 (13.2) | 168.8 (32.4) | 37.3 (1.2) |
| 200 | 647 (10.1) | 91.6 (10.4) | 262.2 (28.1) | **167 (3.2)** |

Table 2: This table presents the results for training on a dataset of 50 thousand transitions but varying the amount of training measured by number of FQI iterations with TTN. The mean of 30 runs is included with one standard error in parentheses.

problem to study. We find that it can be difficult to get good performance empirically with batch RL algorithms, with a common algorithm, DQN, often failing to achieve nontrivial results. On the other hand, in the offline-online setting, the same algorithm can achieve decent performance after a modest amount of training. The batch RL problem may be asking too much of RL algorithms since the agent has to commit to a single fixed policy. Allowing learning afterwards enables quick adaptation to unexpected situations. In practical applications, we are unlikely to be able to collect data for every possible situation, thus it is intuitive to enable the agent to adapt in this manner.

Compared to purely online training, offline-online RL may be more desirable since we can leverage an available dataset to jumpstart the learning process. In a real-world scenario, it may be undesirable to learn purely online since we would like the initial deployed policy to be high-performing, unlike a random policy. Training on an offline dataset is one way to start with reasonable performance and keep improving thereafter.

We hope that this paper generates more interest into offline-online RL. We see this setting as being potentially relevant in many real-world problems where a batch of data is available and could be a more practical solution than either batch or online RL on their own. We see many opportunities to further investigate this setting. From a theoretical perspective, certain work has already started to extend batch RL (Xie et al., 2021), given the unsatisfactory lower bounds (Zanette, 2020). We believe the offline-online RL setting opens the door to many interesting avenues of research.

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

## A  APPENDIX

The appendix is split into two main sections. Additional experiments are found in the first part and experimental details can be found in the second. Within the experimental details, we present the hyperparameters used for all the experiments along with pseudocode for the different settings and algorithms.

### A.1  ADDITIONAL EXPERIMENTS

We repeat the experiments comparing the offline-online setting to batch RL (Fig. 1) and online RL (Fig. 2) with the larger dataset of 50 thousand transitions. The results are qualitatively similar with the larger dataset although the performance of all the algorithms is improved.

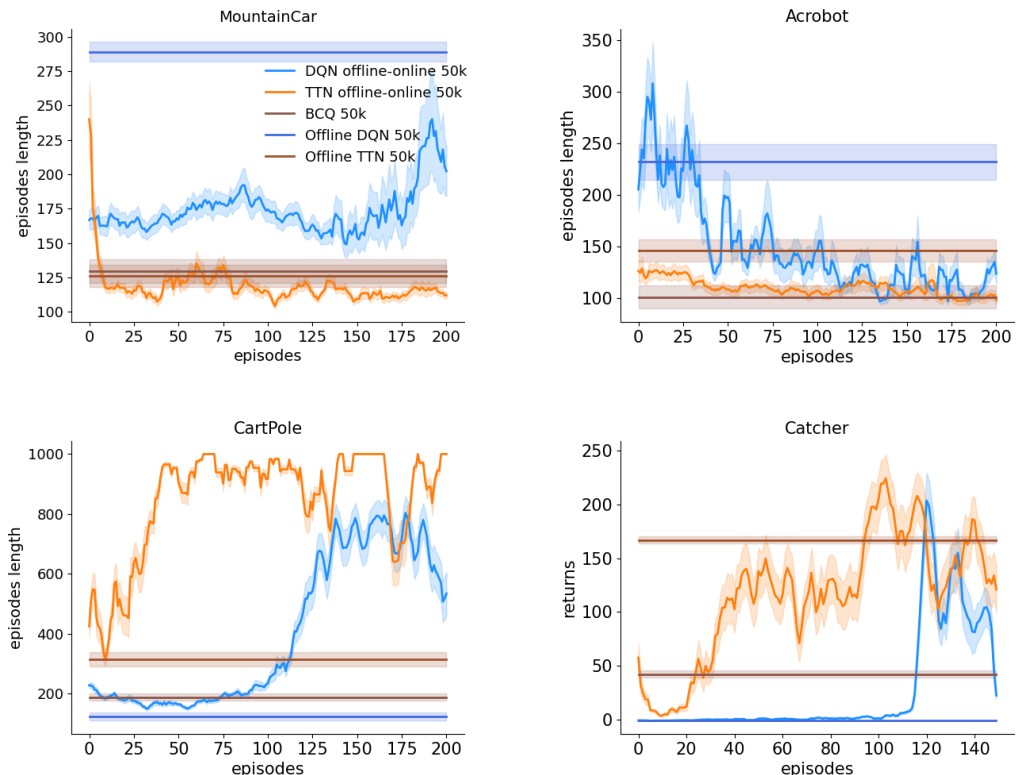

Figure 5: We compare the offline-online setting to batch RL. A dataset of 50 thousand transitions is used.

## A.2 EXPERIMENTAL DETAILS

We present the hyperparameters for all the algorithms and the experiments.

Table 3: This table presents the common hyper-parameters for the TTN and DQN algorithms.

| Parameter | Value |
|---|---|
| Learning rate for TTN | $10^{-3}$ |
| Learning rate for DQN | $10^{-3.75}$ |
| Minibatch size for DQN | 32 |
| Minibatch size for TTN | 32 |
| Minibatch size for offline training | 64 |
| Target estimation in FQI | *Expected SARSA* |
| $\epsilon$ for *Expected SARSA* | 0.01 |

Table 4: This table presents the regularization coefficients and number of offline training for offline-online setting for each environments separately.

| Environments | MountainCar | Acrobot | CartPole | Catcher |
|---|---|---|---|---|
| **Regularization coefficient in FQI** | $10^{-3}$ | $10^{-3}$ | 1 | 1 |
| **Number of offline training for offline-online setting** | 100 | 100 | 70 | 200 |

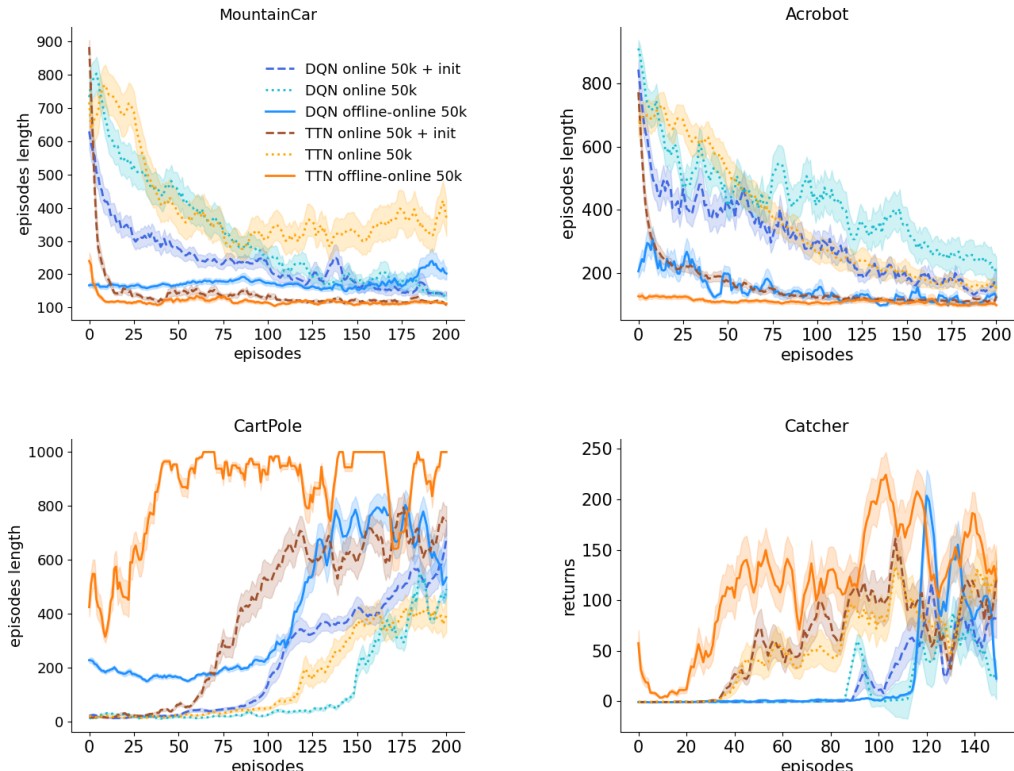

Figure 6: We compare the offline-online setting to purely online learning. We plot learning curves for standard online DQN/TTN, online DQN/TTN initialized with the offline dataset and offline-online DQN/TTN. We use a dataset of 50 thousand transitions here and find qualitatively similar results to using 10 thousand transitions, although performance is usually increased.

To generate the datasets for a good policy, we ran TTN online algorithm for a sufficient number of steps to achieve a good policy and then we stored only the last $K$-steps where $K$ is the size of a dataset. To collect the dataset of 10 thousand transactions, the algortihm was run for 50 thousand steps and for the dataset of 50 thousand transitions, 100 thousands learning steps were used.

We limit each episode of MountainCar, Acrobot, and CartPole to 1000 steps. For Catcher, we cap the (undiscounted) sum of rewards to 300 and terminate the episode if the limit is reached.

### A.2.1 TRAINING PROCEDURES

We include pseudocode for the training procedures of the three settings considered: batch RL, online RL and offline-online RL.

For a generic RL algorithm that has parameters $\theta$ and associated policy $\pi_\theta$, we have the following pseudocode

### A.2.2 ALGORITHMS

We provide pseudocode for TTN and DQN in the different settings here, along with their hyperparameters.

---

**Algorithm 3** Online RL

---

 1: **procedure** ONLINE RL($\theta$)
 2:     Initialize agent parameters $\theta$
 3:     Initialize state $s$ by the environment
 4:     **for** number of episodes **do**
 5:         **while** episode not done **do**
 6:             $a \leftarrow$ action chosen by $\pi_\theta$ given $s$
 7:             $r, s' \leftarrow$ Environment(s, a)                    ▷ Get reward and next state
 8:             Update $\theta$ using $(s, a, r, s')$ (and possibly past samples
 9:             If episode ends, record the total return.
10:         **end while**
11:     **end for**
12: **end procedure**

---

**Algorithm 4** Batch RL

---

 1: **procedure** BATCH RL($\theta, \mathcal{D}$)
 2:     Given dataset $\mathcal{D}$ of transitions $(s, a, r, s')$
 3:     Initialize agent parameters $\theta$
 4:     **for** number of updates **do**
 5:         Update $\theta$ using data from $\mathcal{D}$
 6:     **end for**
 7:     **for** number of evaluation episodes **do**                    ▷ Evaluate the agent
 8:         Initialize state $s$ by the environment
 9:         Run the policy $\pi_\theta$ until episode terminates
10:         Record the return for the episode
11:     **end for**
12: **end procedure**

---

**Algorithm 5** Offline-Online RL

---

 1: **procedure** OFFLINE-ONLINE RL($\theta, \mathcal{D}$)
 2:     Given dataset $\mathcal{D}$ of transitions $(s, a, r, s')$
 3:     Initialize agent parameters $\theta$
 4:     **for** number of updates **do**
 5:         Update $\theta$ using data from $\mathcal{D}$
 6:     **end for**
 7:     **for** number of evaluation episodes **do**                    ▷ Evaluate the agent
 8:         **while** episode not done **do**
 9:             $a \leftarrow$ action chosen by $\pi_\theta$ given $s$
10:             $r, s' \leftarrow$ Environment(s, a)                    ▷ Get reward and next state
11:             Update $\theta$ using $(s, a, r, s')$ (and possibly past samples)  ▷ Allow additional learning
12:             If episode ends, record the total return.
13:         **end while**
14:     **end for**
15: **end procedure**

---

**Algorithm 6** Online TTN

---

1: **procedure** TRAIN( $w, \theta, \hat{w}, \pi$ )
2:     Initialize $\theta, \hat{w}$ with Xavier initialization, $w$ to 0 and starting state $s$ according to the environment.
3:     Initialize data set $D = \Phi$
4:     **for** $N$ number of episodes **do**
5:        $a \leftarrow$ action chosen by $\pi$ given $s$
6:        $r, s' \leftarrow$ Environment(s, a)          ▷ Get reward and next state
7:        $D \leftarrow (s, a, r, s')$          ▷ Add sample to the batch data
8:        $\theta, \hat{w} \leftarrow$ Do SGD on feature learning using sample $(s, a, r, s')$ from mini-batch data $d \in D$
9:        **if** number of steps $== 1000$ **then**
10:          $w \leftarrow$ Do FQI update on value learning using sample $(s, a, r, s')$ from batch data $D$
11:        **end if**
12:     **end for**
13: **end procedure**

---

**Algorithm 7** Offline TTN

---

1: **procedure** TRAIN( $w, \theta, \hat{w}, \pi$ )
2:     Initialize $\theta, \hat{w}$ with Xavier initialization, $w$ to 0, $\tau$ to number of updates, and starting state $s$ according to the environment.
3:     Initialize data set $D$ from a batch of data generated by an arbitrary policy.
4:     **while** $|\theta_{t+1} - \theta_t| \geq \epsilon$ or $t \leq \tau$ **do**          ▷ **Offline-step**
5:        $\theta, \hat{w} \leftarrow$ Do 10000 gradient descent on feature learning using sample $(s, a, r, s')$ from mini-batch data $d \in D$
6:        $w \leftarrow$ Do 1 FQI update on value learning using sample $(s, a, r, s')$ from batch data $D$
7:     **end while**
8:     $t \leftarrow t + 1$
9:     **for** $N$ number of episodes **do**
10:        $a \leftarrow$ action chosen by $\pi$ given $s$
11:        $r, s' \leftarrow$ Environment(s, a)          ▷ Get reward and next state
12:     **end for**
13: **end procedure**

---

**Algorithm 8** Online DQN

---

1: **procedure** TRAIN( $w, \theta, \hat{w}, \pi$ )
2:     Initialize $\theta$ with Xavier initialization, $w$ to 0 and starting state $s$ according to the environment.
3:     Initialize data set $D = \Phi$
4:     **for** $N$ number of episodes **do**
5:        $a \leftarrow$ action chosen by $\pi$ given $s$
6:        $r, s' \leftarrow$ Environment(s, a)          ▷ Get reward and next state
7:        $D \leftarrow (s, a, r, s')$          ▷ Add sample to the batch data
8:        $\theta, \hat{w} \leftarrow$ Do SGD on feature learning using sample $(s, r, s')$ from mini-batch data $d \in D$
9:        **if** number of steps $= 1000$ **then**
10:          $w \leftarrow$ Do 1 FQI update on value learning using sample $(s, r, s')$ from batch data $D$
11:        **end if**
12:     **end for**
13: **end procedure**

---

