# OpenReview forum: "Offline-Online Reinforcement Learning: Extending Batch and Online RL"
_ICLR.cc/2022/Conference — ICLR 2022 Submitted_

### Official Review · Reviewer_Vqyy · 2021-11-01

**Correctness:** 3
**Technical Novelty And Significance:** 1
**Empirical Novelty And Significance:** 2
**Recommendation:** 3
**Confidence:** 3

**Main Review:**

Strength:
- The paper studies an important setting that combines online and offline RL, which seems more closely related to some real-world applications, while typically a large amount of historical data exists and could be used properly to warm start the algorithm.
- The experiments using DQN and TTN to successfully illustrate the effectiveness of this combined setting, compared with purely online/offline setting, though there is some minor problem for a fair comparison.
- It is interesting to see learning a representation layer separately is better than the end-to-end learning.

Weakness:
- This online/offline setting is well-studied in the literature, such as [1],[2] pointed in the above section. It would be great if the authors could compare with them explicitly.
- This is an empirical paper, however the take-away from the experiments is trivial and well-known to the community. As accessing more online data should be better than purely offline learning, and I do not see any novelty or meaningful takeaway here.
- The experiments are based on DQN and TTN in both offline and online cases. DQN is suffered from extrapolation error, and there are new algorithms to handle this, such as MOPO, CQL, etc. It is interesting to see the offline learning part based on some of these STOA algorithms, though a similar conclusion might hold there.
- There is also some problem in the experiment design, as it is unfair for the offline learning algorithm, it should access the same batch data, as well as the similar quality online exploration data.

**Summary Of The Paper:**

This paper studies the offline-online setting in RL, in which the agent is given a batch of offline data to train and could further refine it with exploration with the environment.

By testing on two specific algorithms, DQN and TTN, this paper shows empirically that (1). compared with a purely offline setting, it is beneficial to allow the policy to adapt online and (2). compared with the purely online setting, it is beneficial to use the offline/batch data to warm start. Beyond this, they also empirically verify how the amount and quality of offline data that is used to pre-train the policy help later adaptation stage.

The offline-online setting is interesting, but it is not novel, as there are some work studies this, such as [1], [2]. The empirical experiments are valuable but too specific to the particular algorithms, DQN and TTN. The conclusion from the experiments is trivial and well-known to the community as well.


[1]. Policy Finetuning: Bridging Sample-Efficient Offline and Online Reinforcement Learning
[2]. QT-Opt: Scalable Deep Reinforcement Learning for Vision-Based Robotic Manipulation


**Summary Of The Review:**

Overall, though the setting this paper studied is interesting. The experiments do not provide any new insight into this setting and I would recommend a reject.

---

### Official Review · Reviewer_FHnd · 2021-11-03

**Correctness:** 4
**Technical Novelty And Significance:** 1
**Empirical Novelty And Significance:** 2
**Recommendation:** 3
**Confidence:** 5

**Main Review:**

**The setting considered in this paper is not new.** As the authors also acknowledge, prior works such as Yu et al. 2020 have a similar setting. For example, Yu et al. 2020 run a random policy to collect a dataset. It then learns representations of the world and later uses them during online exploration and policy training. In prior works on online RL, replay buffers are used during training which is another related setting. The exact setting of this paper (policy fine-tuning) is presented by Xie et al. 2021.

**The message of the paper is not new.** The messages presented in this paper are:
- If we train a policy using an offline dataset and fine-tune it with additional online data, we get a better policy than only using the offline dataset.
- If we train a policy using an offline dataset and fine-tune it with additional online data, we get a better policy compared to if we start from a random initial policy and fine-tune it with online data of the same budget.

Such improvements are expected to be observed in empirical results and the level of improvement depends on the quality of the dataset. Investigating this setting from a theoretical perspective (as done by Xie et al. 2021) provides insights as in theory, we are concerned with questions such as exact convergence rates, how many additional samples are needed during fine-tuning, what guarantees does the overall offline-online setting provide, etc. I don't see any new insights provided by the empirical studies in this paper.

**The empirical comparisons are not fair.**
- In Figure 1, offline-only RL and offline-online RL are compared. However, the offline-only RL is provided with strictly fewer samples and therefore this comparison is not fair.
- In Figure 2, online-only and offline-online RL are compared. Again, offline-online RL is provided with a dataset in addition to the same online exploration budget and therefore this comparison is not fair.
- Even if the sample sizes were the same, the comparison between these settings highly depends on the quality and type of dataset (e.g. does it come from policy rollouts? is it random? is it adversarial?) and I'm not sure how one can fairly compare these settings.

Question: How are the batch datasets collected in the experiments?

**Summary Of The Paper:**

This paper focuses on reinforcement learning, where the agent is given an offline dataset and is also allowed to collect additional data during online exploration. This allows the agent to finetune the policy and adapt to new situations. The authors run experiments showing that the offline-online RL outperforms online only or offline only RL.

**Summary Of The Review:**

Overall the paper has limited novelty and new insights.

---

### Official Review · Reviewer_NS43 · 2021-11-04

**Correctness:** 3
**Technical Novelty And Significance:** 1
**Empirical Novelty And Significance:** 2
**Recommendation:** 5
**Confidence:** 3

**Main Review:**

Major Comments:

I think the authors consider an important and practical setting of offline-online RL where the algorithm has access to both offline data and a limited amount of online interactions. However, I am concerned with the novelty of the paper. Offline-online RL has been considered before (for example, AWAC [1]), and to my knowledge, the authors do not discuss a comparison of their work to existing literature. The proposed algorithm can be interpreted as training on the offline dataset and fine-tuning online, which [1] also does just using a different base RL algorithm. Furthermore, the authors only compare their method to RL algorithms that learn solely offline or online, so it is unsurprising that learning on both improves performance. It would be useful if the authors also compared their algorithm to existing offline-online RL algorithms.


Minor Comments:

It is not completely clear to me what the difference between offline-online TTN and offline-online DQN are. My hypothesis is that DQN does not fix the representation when learning the value, but that is not clear from the paper.

In Algorithm 1, it would be more clear if the operation showed (s, a, r, s’) as being appended to dataset D. The comment makes this clear, but the actual pseudocode makes it look like a straightforward assignment.

[1] https://arxiv.org/pdf/2006.09359.pdf


**Summary Of The Paper:**

This paper proposes an offline-online RL algorithm that learns jointly on offline data and online interactions. The proposed algorithm, offline-online TTN, learns by separately learning a representation, and values given a fixed representation, on two different timescales. The authors evaluate their algorithm against offline-online DQN, and TNN and DQN that only learns using offline data or online.

**Summary Of The Review:**

Overall, I feel that the novel components of this paper compared to existing work in offline-online RL is not clear. Hence, I am recommending that the paper be rejected. However, if the authors provide a clear discussion and empirical comparison to prior work (AWAC, for example), I would be willing to raise my score.

---

### Official Review · Reviewer_aP1R · 2021-11-08

**Correctness:** 4
**Technical Novelty And Significance:** 3
**Empirical Novelty And Significance:** 3
**Recommendation:** 5
**Confidence:** 3

**Main Review:**

The setting introduced in this paper is interesting. It combines the online setting and the offline setting which allows the agent to achieve better performance than both settings.

However, I have a few concerns that prevent  me from recommending an acceptance.

(i) The proposed algorithm is a simple modification to an existing algorithm (Two-Timescale Networks), and therefore the algorithmic contribution is limited.

(ii) The empirical observations are not supported by theoretical analysis. Also, the experiments are performed only on a small number of relative simple environments (acrobot, cartpole, etc). It is unclear if the same trend holds for other environments (Atari games, Mujoco, etc).

(iii) In the experiments, the algorithm is effective only when the batch dataset is collected from a policy which performs well, and if we use random policy to collect the batch dataset, the algorithm cannot achieve non-trivial performance. However, with samples from a good policy, it is not surprising to me that the algorithm performs better than those in the online setting since those samples could help warm start the training process. The authors could consider performing experiments on other batch dataset. E.g., what will happen if the dataset is collected by a policy with middle-level performance? Such a policy can be obtained by early stopping online RL algorithms.

Due to these concerns, I cannot recommend an acceptance at this point, and I am open to raise my score if the authors could address my concerns.

**Summary Of The Paper:**

In this paper, the authors explore the offline-online setting for RL, where the agent has access to a batch of data to train on but is also allowed to learn during the evaluation phase in an online manner. Compared to the offline setting, the agent is now allowed to take online samples during the evaluation phase. Compared to the online setting, the agent has access to a batch dataset which could potentially improve the performance.

The authors demonstrate that empirically, an offline-online variant of Two-Timescale Networks outperforms online RL / offline RL algorithms. The authors also perform ablation study to investigate factors that affect the performance.

**Summary Of The Review:**

See above.

---

### Decision · Program_Chairs · 2022-01-20

**Decision:**

Reject

**Comment:**

It seems that the reviewers reached out a consensus that the paper is not ready for publication at ICLR. The reviewers raised concerns including “The empirical observations are not supported by theoretical analysis” , “The proposed algorithm is a simple modification to an existing algorithm”, concerns with “with the novelty of the paper”, “The message of the paper is not new. “ Please see the reviews for more detailed discussions about the paper.